# Gestational diabetes mellitus and offspring's carotid intima–media thickness at birth: MySweetHeart Cohort study

Adina Mihaela Epure ●,[1,2] Stefano Di Bernardo ●,[3] Yvan Mivelaz ●,[3] Sandrine Estoppey Younes,[2] Arnaud Chiolero ●,[1,4,5] Nicole Sekarski ●,[3] On behalf of MySweetHeart Research Group

AC and NS contributed equally.

For numbered affiliations see end of article.

**Correspondence to**
Professor Nicole Sekarski;
nicole.sekarski@chuv.ch

## ABSTRACT

**Objective** Hyperglycaemia during pregnancy is associated with cardiometabolic risks for the mother and the offspring. Mothers with gestational diabetes mellitus (GDM) have signs of subclinical atherosclerosis, including increased carotid intima–media thickness (CIMT). We assessed whether GDM is associated with increased CIMT in the offspring at birth.

**Design and setting** MySweetHeart Cohort is a prospective cohort study conducted in Switzerland.

**Participants, exposure and outcome measures** This work included pregnant women with and without GDM at 24–32 weeks of gestation and their singleton live-born offspring with data on the primary outcome of CIMT. GDM was diagnosed based on the criteria of the International Association of Diabetes and Pregnancy Study Groups. Offspring's CIMT was measured by ultrasonography after birth (range 1–19 days).

**Results** Data on CIMT were available for 99 offspring of women without GDM and 101 offspring of women with GDM. Maternal age ranged from 18 to 47 years. Some 16% of women with GDM and 6% of women without GDM were obese. Smoking during pregnancy was more frequent among women with GDM (18%) than among those without GDM (4%). Neonatal characteristics were comparable between the two groups. The difference in CIMT between offspring of women with and without GDM was of 0.00 mm (95% CI −0.01 to 0.01; p=0.96) and remained similar on adjustment for potential confounding factors, such as maternal prepregnancy body mass index, maternal education, smoking during pregnancy, family history of diabetes, as well as offspring's sex, age, and body surface area (0.00 mm (95% CI −0.02 to 0.01; p=0.45)).

**Conclusions** We found no evidence of increased CIMT in neonates exposed to GDM. A longer-term follow-up that includes additional vascular measures, such as endothelial function or arterial stiffness, may shed further light on the cardiovascular health trajectories in children born to mothers with GDM.

**Trial registration number** NCT02872974; Pre-results.

## STRENGTHS AND LIMITATIONS OF THIS STUDY

⇒ One important strength of this study is represented by its prospective design and the enrolment of participants at the time of gestational diabetes diagnosis.
⇒ Carotid intima–media thickness was measured in non-sedated neonates by experienced paediatric cardiologists using automated methods with manual tracing adjustment, in accordance with published guidelines.
⇒ Limitations of this study include the relatively small sample size, the possibility of residual confounding and the limited generalisability.

recognition during pregnancy.[1–3] The prevalence of hyperglycaemia during pregnancy has increased in recent decades, being estimated at 16% worldwide in 2019, with 84% of cases due to GDM.[4] GDM is associated with long-term metabolic consequences for both the mother and the offspring, such as type 2 diabetes and obesity.[5] Women with GDM also have subclinical atherosclerosis and an increased risk for cardiovascular disease (CVD) later in life.[6 7] However, little is known about the cardiovascular risk of their offspring.

Carotid intima–media thickness (CIMT) is a surrogate marker of atherosclerosis, which has been shown to be increased in children exposed to risk factors in the first 1000 days of life, such as poor fetal growth,[8] as well as in children with type 1 diabetes.[9] From a developmental origins of health and disease perspective,[10] exposure to adverse experiences in early life may produce lifelong adaptations in the organs' structure and function and may programme the risk for CVD. For instance, a systematic review and meta-analysis showed that GDM was associated with a higher systolic blood pressure in childhood.[11]

## INTRODUCTION

Gestational diabetes mellitus (GDM) is a state of hyperglycaemia with onset or first

Whether GDM has an impact on children's CIMT is not clearly established. The evidence is scarce notably in the very young children although CIMT measurement is feasible from birth and could help discern between changes that occur before or after birth.[12] To fill this gap, we conducted MySweetHeart Cohort study to assess the early life cardiovascular consequences of GDM.[13] Here, we evaluated CIMT at birth in offspring of mothers with and without GDM.

## METHODS

### Study design and setting

MySweetHeart Cohort is a prospective cohort study conducted at the Lausanne University Hospital (CHUV), Switzerland.[13]

### Study population

This cohort included pregnant women between 24 and 32 weeks of gestation, with and without GDM. Other inclusion criteria were age 18 years or more and understanding French or English. The exclusion criteria were pre-existing diabetes mellitus, strict bed rest or severe mental disorders. To facilitate recruitment and share resources, a collaboration was established with MySweetHeart Trial (NCT02890693),[14] a randomised controlled trial assessing the effect of a lifestyle and psychosocial intervention on cardiometabolic outcomes of women with GDM and their offspring. As such, women with GDM were invited to contribute to both studies. Participating women with and without GDM were included in the current analysis if CIMT data for their live-born singleton neonates were available. All families gave a signed informed consent for use of their data.

### Data collection

#### GDM screening

Pregnant women screened at the prenatal care clinic of the CHUV had a fasting plasma glucose (FPG) test between 24 and 28 weeks of gestation and GDM was diagnosed if the test result was ≥5.1 mmol/L.[13] If FPG was <5.1 mmol/L, but ≥4.4 mmol/L, women had a 2-hour 75 g oral glucose tolerance test (OGTT) and GDM was diagnosed based on the criteria of the International Association of Diabetes and Pregnancy Study Groups (IADPSG).[15] Pregnant women screened by external obstetricians in the Canton of Vaud underwent the same procedure or directly a 2-hour 75 g OGTT.

#### Carotid ultrasound and CIMT measurement

A carotid ultrasound assessment was performed between 1 and 7 days of life in the majority of neonates (n=191). A small share (n=9) had the exam between 8 and 19 days of life due to organisational and logistical constraints. Parents were told to feed and burp their offspring ahead of the carotid ultrasound to make them more relaxed. Feeding or administration of a 30% glucose solution were used to comfort the neonates if they became agitated

during the exam. The exam took place in a dark and quiet room and a cloth was placed under the neonates' shoulders to facilitate the extension of the neck.

Ultrasound image acquisition and analysis were performed by two experienced paediatric cardiologists who were blinded to the maternal glycaemic status. Images were acquired in B-mode with no harmonics, sonoCT, dynamic range of 60 dB, at a frame rate of 100–120 Hz, with a depth of 1–2 cm. The right and left carotid arteries were scanned using a Philips EPIC echocardiograph (Philips Medical, Netherlands) with a L 15–7 MHz high-resolution linear array transducer, according to the American Heart Association's recommendations for standard assessment of subclinical atherosclerosis in children and adolescents.[16] Each observer recorded three consecutive 3 s cine loops from two different angles on each side, which were stored as native Digital Imaging and Communications in Medicine (DICOM) format for subsequent offline analyses (QLab, Philips Medical, Netherlands). Whenever image quality was optimal enough, six right and six left frames were selected and, for each, the maximal intima-media thickness of the common carotid artery far wall was measured. Measurements were performed over a 1 cm region of interest proximal to the carotid bulb, on or closest to the R-wave of the ECG, using a semiautomated edge detection software with manual tracing adjustment when needed. The mean of 12 maximal CIMT measurements was used in the analysis for the majority of neonates (n=170). Two neonates had only one measurement available, whereas the rest had between 2 and 11 measurements that were averaged. A good interobserver reliability (coefficient of variation=5.9%) for measurements in non-sedated infants was proven in our laboratory previously.[12]

### Other sample characteristics

Data on maternal characteristics (age, country of origin, education, smoking during pregnancy, prepregnancy weight and height, or parity) and family history of diabetes were record-based or self-reported by the mother at a researcher-administered interview on inclusion in the study. Smoking during pregnancy was defined as a mother who was an active tobacco smoker at study baseline, that is, between 24 and 32 weeks of gestation. A maternal blood sampling was also performed at baseline and glycated haemoglobin was measured. Prepregnancy body mass index (BMI) was computed by dividing the prepregnancy weight (kg) by the squared height ($m^2$). Delivery data such as newborn sex, anthropometry, gestational age or mode of delivery were obtained from the medical records. Neonatal weight, length and blood pressure were measured by the study team at the time of the carotid ultrasound. Body surface area ($m^2$) was computed using the Mosteller equation.[17] One systolic and diastolic blood pressure measurement was taken from the right upper arm, in a supine position, using a clinically validated and regularly calibrated oscillometric

sphygmomanometer (Accutorr Plus; Datascope, Paramus, New Jersey, USA) with neonate cuffs.

## Data analysis

Descriptive statistics on study participants are reported as percentages (%) or as mean, standard deviation (SD), minimum and maximum values. The relationship of GDM with CIMT was evaluated by a set of linear regression models with and without adjustment for potential confounders, that is, baseline covariates associated with metabolic and cardiovascular risks, offspring's sex and anthropometry at CIMT assessment. Potential confounders were maternal prepregnancy BMI, maternal education (university/no university), smoking during pregnancy (yes/no) and family history of diabetes (yes/no). The variable family history of diabetes summarised disease occurrence in a first-degree relative of the mother, first-degree relative of the father or in the father himself and assumed missing data in any of these variables as no history of diabetes unless values for all three variables were missing. To account for differences in body size,[18 19] we adjusted for body surface area and age at CIMT assessment. All statistical analyses were performed in Stata V.16.

## Patient and public involvement

Patients and/or the public were not involved in the design, or conduct, or reporting, or dissemination plans of this research.

## RESULTS

### Characteristics of study participants

Data collection started in September 2016 and ended in October 2020. A total of 137 participants without GDM exposure and 212 participants with GDM exposure were recruited in the study. Some 101 neonates without GDM exposure and 117 neonates with GDM exposure attended the cardiovascular follow-up visit early after birth. Of these, 200 singleton neonates born at more than 36 weeks of gestation (non-GDM: n=99; GDM: n=101) had CIMT measurements and constitute the analytical sample for the current analysis.

**Table 1** Characteristics of study participants by GDM exposure

| | Non-GDM* (n=99) | | | | GDM† (n=101) | | | |
|---|---|---|---|---|---|---|---|---|
| | Mean or % | SD | Min | Max | Mean or % | SD | Min | Max |
| **Maternal** | | | | | | | | |
| Age (years) | 33 | 5 | 18 | 44 | 33 | 5 | 21 | 47 |
| Swiss origin (%) | 24 | | | | 33 | | | |
| University education (%) | 60 | | | | 55 | | | |
| Primiparous (%) | 55 | | | | 48 | | | |
| Smoking during pregnancy (%) | 4 | | | | 18 | | | |
| Prepregnancy obesity (BMI ≥30 kg/m$^2$) (%) | 6 | | | | 16 | | | |
| HbA1c (%) | 4.9 | 0.3 | 4.2 | 5.7 | 5.3 | 0.3 | 4.7 | 7.2 |
| **Neonatal** | | | | | | | | |
| Male (%) | 52 | | | | 53 | | | |
| Caesarean section (%) | 22 | | | | 32 | | | |
| Term birth (37–41 weeks) (%) | 98 | | | | 96 | | | |
| Birth weight (g) | 3352 | 425 | 2190 | 4190 | 3357 | 442 | 2220 | 4340 |
| Macrosomia (birth weight >4000 g) (%) | 5 | | | | 6 | | | |
| Length (cm) | 50 | 2 | 45 | 54 | 50 | 2 | 45 | 56 |
| Body surface area (m$^2$) | 0.21 | 0.02 | 0.16 | 0.25 | 0.21 | 0.02 | 0.17 | 0.26 |
| Systolic BP (mm Hg) | 78 | 9 | 60 | 101 | 78 | 10 | 60 | 111 |
| Diastolic BP (mm Hg) | 47 | 8 | 30 | 66 | 48 | 10 | 28 | 90 |
| Family history of diabetes (%) | 24 | | | | 46 | | | |

*Non-GDM: missing values for Swiss origin (n=1), university education (n=2), pre-pregnancy obesity (n=1), HbA1c (n=13), caesarean section (n=4), term birth (n=10), systolic BP (n=1), diastolic BP (n=1); family history of diabetes (n=1).
†GDM: missing values for age (n=3), Swiss origin (n=3), university education (n=18), primiparous (n=3), smoking (n=5), prepregnancy obesity (n=4), HbA1c (n=5), male (n=16), caesarean section (n=6), term birth (n=16), birth weight (n=16); family history of diabetes (n=4).
BMI, body mass index; BP, blood pressure; GDM, gestational diabetes mellitus; HbA1c, glycated haemoglobin; n, number of participants; SD, standard deviation.

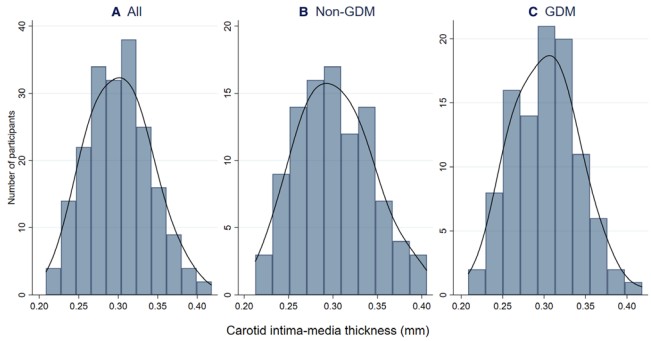

**Figure 1** Histograms of CIMT at birth, overall and by GDM exposure. This figure shows the distribution of CIMT values in our sample, overall (n=200) and by GDM exposure (non-GDM: n=99; GDM: n=101). The black line represents the kernel density estimate. CIMT, carotid intima–media thickness; GDM, gestational diabetes mellitus; n, number of participants.

Family and neonatal characteristics of study participants are presented in table 1. The maternal characteristics were generally comparable between the non-GDM and GDM groups. The majority of women were non-Swiss and their age ranged from 18 to 47 years. Approximately half of the women in each group had a high level of education and no previous deliveries. More women with GDM (16%) were obese (prepregnancy BMI ≥30 kg/m$^2$) compared with women without GDM (6%). Smoking during pregnancy was more frequent among women with GDM (18%) than among those without GDM (4%). Offspring of women with and without GDM had similar neonatal characteristics, such as sex, gestational age, birth weight, length or blood pressure. The majority were born at term, that is, between 37 and 41 weeks (GDM: 96%; non-GDM: 98%) and a small share had macrosomia, that is, a birth weight higher than 4'000 g (GDM: 6%; non-GDM: 5%). Offspring of women with GDM (46%) had a higher frequency of family history of diabetes compared with their non-GDM counterparts (24%).

### GDM and CIMT at birth

The distribution of CIMT values is presented in figure 1, figure 2 and online supplemental figure S2. CIMT ranged from 0.21 to 0.42 mm, with a mean CIMT of 0.30 mm (SD 0.04) overall and in each of the studied groups (table 2, online supplemental table S1).

The relationship of GDM with offspring's CIMT early after birth is presented in table 2 and figure 3. In the unadjusted analysis (model 1), the difference in CIMT between offspring of women with and without GDM was 0.00 mm (95% CI –0.01 to 0.01; p=0.96). Adjustment for offspring sex and potential confounding factors (model 2), as well as for offspring's body surface area and age at CIMT assessment (model 3), resulted in a difference of 0.00 mm (95% CI –0.02 to 0.01; p=0.45). When exposure to GDM was analysed separately for offspring whose mothers were assigned or not to a lifestyle and psychosocial intervention as part of their participation in MySweetHeart Trial,

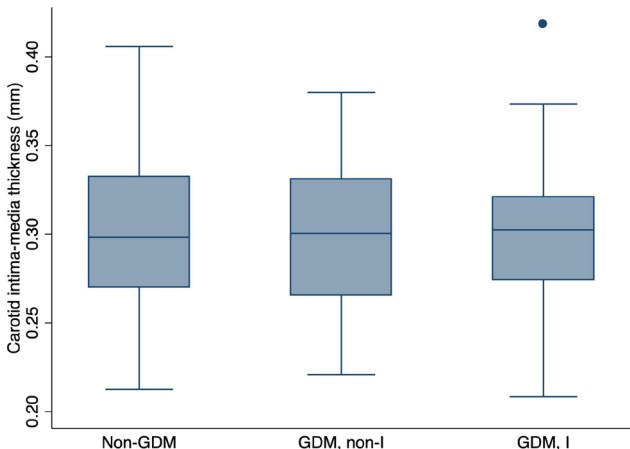

**Figure 2** Box plots of CIMT at birth by GDM exposure and assignment to a lifestyle and psychosocial intervention. This figure shows the distribution of CIMT in the offspring of women without GDM (non-GDM; n=99) and the offspring of women with GDM who were assigned to no intervention (GDM, non-I; n=48) or to a lifestyle and psychosocial intervention (GDM, I; n=53) as part of their participation in the MySweetHeart trial. The line inside the box represents the median value of the distribution, while the lower and upper boundaries of the box represent the first (Q1) and third (Q3) quartiles, respectively. The IQR corresponds to Q3–Q1. The whiskers extend from either side of the box up to 1.5*IQR (ie, Q1–1.5*IQR and Q3 +1.5*IQR). Outliers are depicted as circles. CIMT, carotid intima–media thickness; GDM, gestational diabetes mellitus; I, intervention; IQR, interquartile range; n, number of participants.

results were similar to those presented above (online supplemental table S1 and online supplemental figure S1).

### DISCUSSION
#### Summary of findings and comparison with other studies
Our goal was to assess the relationship of GDM with neonatal CIMT. We found no evidence of an increased CIMT in neonates born to women with GDM as compared with those born to women without GDM. Our findings are in line with other studies that evaluated CIMT after intrauterine exposure to maternal hyperglycaema. A recent meta-analysis pooled data from three studies and reported no clear evidence of increased CIMT in children exposed to maternal hyperglycaemic compared with those not exposed (pooled standardised mean difference (SMD): 0.08 (95% CI –0.16 to 0.33)).[8] Two of these studies included 6-year and 8-year children, respectively, and found no difference in CIMT after exposure to GDM (SMD 0.00 (95% CI –0.28 to 0.28) at 6 years and 0.00 (95% CI –0.41 to 0.41) at 8 years).[8 20 21] The third study included neonates and found a slightly higher CIMT among those exposed to diabetes (SMD 0.46 (95% CI –0.07 to 1.00)),[8 22] but the imprecision around the estimated difference was high, the study had a very small sample size (n=55) and the authors did not specify

**Table 2** The relationship of GDM with offspring's CIMT at birth

| | Mean (SD), mm | Model 1 (n=200) | | Model 2 (n=165) | | Model 3 (n=165) | |
|---|---|---|---|---|---|---|---|
| | | Mean difference (95% CI), mm | P value | Mean difference (95% CI), mm | P value | Mean difference (95% CI), mm | P value |
| Non-GDM | 0.30 (0.04) | Ref | Ref | Ref | Ref | Ref | Ref |
| GDM | 0.30 (0.04) | 0.00 (–0.01 to 0.01) | 0.96 | 0.00 (–0.02 to 0.01) | 0.47 | 0.00 (–0.02 to 0.01) | 0.45 |

Estimates were obtained from linear regression models with the following specification: Model 1: unadjusted estimates; Model 2: estimates adjusted for maternal prepregnancy BMI, education and tobacco smoking; offspring family history of diabetes and sex; Model 3: estimates adjusted for maternal prepregnancy BMI, education and tobacco smoking; offspring family history of diabetes, sex, body surface area and age at CIMT assessment. The outcome variable (ie, CIMT) was continuous. The exposure variable was binary (GDM/non-GDM; the reference category was non-GDM). Similar results were obtained when Model 1 was run in the sample (n=165) with data on outcome, exposure and all covariates included in Model 2 and Model 3 (GDM: 0.00 mm (95% CI –0.02 to 0.01; p=0.54)).
BMI, body mass index; CI, confidence interval; CIMT, carotid intima–media thickness; GDM, gestational diabetes mellitus; n, number of participants; Ref, reference group; SD, standard deviation.

whether they included women with pregestational or gestational diabetes.[22]

### Strengths and limitations

A major strength of this study is its prospective design. Enrolment of study participants and collection of baseline characteristics took place close to the moment of GDM diagnosis and ahead of the CIMT outcome assessment. This implies that the choice of participation in the study is unlikely to be related to both the exposure and the outcome, which makes selection bias due to enrolment unlikely. Further, GDM was diagnosed using the new criteria of the IADPSG. These criteria were derived based on the risk of adverse neonatal outcomes, such as birth weight, cord blood C-peptide levels or per cent infant body fat >90th percentile.[15] They were endorsed by the World Health Organization (WHO) along with several

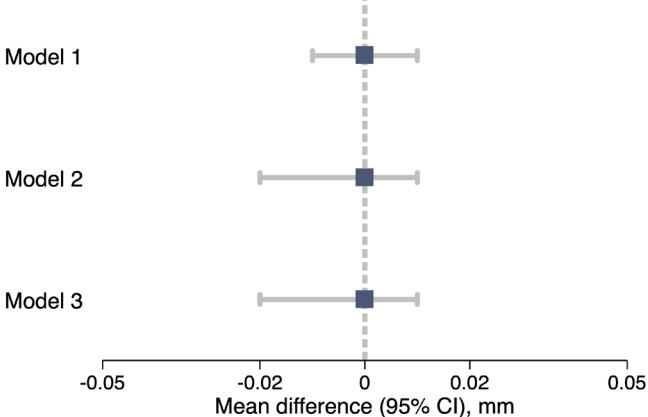

**Figure 3** Illustration of the relationship of GDM with offspring's CIMT at birth through a forest plot. The boxes represent the mean differences in CIMT between offspring of women with and without GDM (ie, GDM vs non-GDM). The horizontal lines represent the 95% CIs. The plot was constructed using regression estimates and models presented in table 2. Model specification: Model 1 is unadjusted, while Model 2 and Model 3 are adjusted for various factors as described in the methods and footnote of table 2. CI, confidence interval; CIMT, carotid intima–media thickness; GDM, gestational diabetes mellitus.

other bodies to achieve a universal consensus for GDM diagnosis and increase comparability of the evidence.[23 24] Another strength is the assessment of ultrasound CIMT using automated methods with manual tracing adjustment, in accordance with the current guidelines in children.[16 25] The semiautomated methods are associated with a lower interoperator variability and high reliability,[16 25] including in infants, as it was previously proved in our laboratory.[12]

This study has some limitations. First, our results have limited generalisability, as we used a convenient sample of pregnant women recruited from healthcare facilities in Switzerland. Second, the GDM glucose screening test (FPG or 75 g OGTT) varied between participants. This is because our hospital used a two-step targeted approach for identifying women with GDM. While the two-step approach is practical and more acceptable to patients,[26] it may be related to a lower likelihood of diagnosing GDM compared with a one-step universal screening based on a 75 g OGTT.[27] On the other hand, the IADGSP criteria, which have a lower threshold for a positive FPG test (≥5.1 mmol/L) compared with other guidelines,[23] may identify as having GDM women who are at low absolute risk for fetal and pregnancy complications and, thus, overdiagnose GDM in some populations.[28 29] Therefore, misclassification of the exposure cannot be excluded and our estimates of association might be biased, maybe underestimated. Additionally, women with GDM participated in MySweetHeart Trial and approximately half of them were assigned to a lifestyle and psychosocial intervention with the aim of improving their cardiometabolic outcomes. Although this intervention could have also modified the association of GDM with CIMT, this seems not likely, as mean CIMT values were very similar in offspring of women with GDM who participated in the intervention and the control arms of the trial. Third, CIMT was assessed using conventional high-resolution ultrasound frequencies (<15 MHz), which have a lower spatial resolution and, thus, tend to overestimate the arterial thickness in the young children when compared with very high-resolution ultrasound systems (25–55 MHz).[30 31] Measurement error

in CIMT cannot be excluded, but systematic differences between the two groups are unlikely because the outcome assessors were blinded to the glycaemic status of the mothers. Fourth, while we adjusted for key confounders at the analysis stage, there is a possibility of bias due to unmeasured factors, such as family history of premature cardiovascular death, or residual confounding due to the relatively small sample size and imprecision in the measurement of confounder variables, especially in those self-reported. Last, our study was limited to CIMT, which is a measure of arterial structure. In fact, changes in the vessel function might occur earlier than changes in the vessel structure, therefore, a combination of vascular measures would be needed for a clearer view on the cardiovascular status of children exposed to adverse experiences in early life. However, certain techniques to assess arterial function and stiffness, such as flow-mediated dilation and pulse-wave velocity, are not currently feasible in the very young due to limited compliance and technical inconveniences.[18]

## Implications and future research

Our results suggest that intrauterine exposure to GDM does not induce changes in the carotid artery structure that are detectable with conventional ultrasound techniques at birth and may not be linked to early vascular ageing at this arterial site in the short term. Measurements at other arterial sites, such as the aorta,[32] may be more useful to investigate early or subtle abnormalities related to accelerated vascular ageing or subclinical atherosclerosis. A long-term follow-up that includes complementary vascular measures, for instance, endothelium-dependent and endothelium-independent vasodilation or large-artery stiffness,[20] may shed further light on the cardiovascular health of children born to mothers with GDM.

**Author affiliations**
[1]Population Health Laboratory (#PopHealthLab), University of Fribourg, Fribourg, Switzerland
[2]Department of Epidemiology and Health Services, Center for Primary Care and Public Health (UNISANTÉ), University of Lausanne, Lausanne, Switzerland
[3]Paediatric Cardiology Unit, Woman-Mother-Child Department, Lausanne University Hospital (CHUV), Lausanne, Switzerland
[4]Institute of Primary Health Care (BIHAM), University of Bern, Bern, Switzerland
[5]School of Global and Population Health, McGill University, Montréal, Québec, Canada

**Acknowledgements** The study teams of MySweetHeart Cohort and MySweetHeart Trial form MySweetHeart Research group and collaborated on data collection, management, and curation. We thank the members of MySweetHeart Research Group, listed in alphabetical order under Collaborators.

**Collaborators** Collaborator group name: MySweetHeart Research Group. Individual names listed in alphabetical order: Amar Arhab, Pascal Bovet, Arnaud Chiolero, Stefano Di Bernardo, Adina Mihaela Epure, Sandrine Estoppey Younes, Leah Gilbert, Justine Gross, Antje Horsch, Stefano Lanzi, Seyda Mayerat, Yvan Mivelaz, Jardena J. Puder, Dan Quansah, Jean-Benoit Rossel, Nicole Sekarski, Umberto Simeoni, Bobby Stuijfzand, Yvan Vial.

**Contributors** AC, NS, SDB and YM designed the study and the data collection procedures with input from SEY, AME. SEY and AME collected baseline characteristics for participants without GDM. SDB and NS collected neonatal cardiovascular characteristics for all participants. SEY performed data management and curation. AME carried out the statistical analyses with input and supervision from AC. AME wrote the first draft of the manuscript with input from AC and NS. SDB, SEY, YM made critical revisions to the manuscript for important intellectual content. All authors read and approved the content of the manuscript. NS and AC act as guarantors and take primary responsibility for the final content.

**Funding** This study was funded by the Swiss National Science Foundation (www.snf.ch; MySweetHeart Cohort project number 32003B-163240; MySweetHeart Trial project number 32003B_176119).

**Competing interests** None declared.

**Patient and public involvement** Patients and/or the public were not involved in the design, or conduct, or reporting, or dissemination plans of this research.

**Patient consent for publication** Not applicable.

**Ethics approval** This study involves human participants and was approved by the Ethics Committee for Human Research of the Canton of Vaud (ID 2016-00745). Participants gave informed consent to participate in the study before taking part.

**Provenance and peer review** Not commissioned; externally peer reviewed.

**Data availability statement** Data are available on reasonable request. Data could be made available by the principal investigator and corresponding author (NS: nicole.sekarski@chuv.ch) on reasonable request.

**ORCID iDs**
Adina Mihaela Epure http://orcid.org/0000-0002-8104-0244
Stefano Di Bernardo http://orcid.org/0000-0003-1977-3727
Yvan Mivelaz http://orcid.org/0000-0002-3507-6879
Arnaud Chiolero http://orcid.org/0000-0002-5544-8510
Nicole Sekarski http://orcid.org/0000-0002-3269-249X

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
