## [Reviewer comments · BMJ Open]

ARTICLE DETAILS

TITLE (PROVISIONAL)	Gestational diabetes mellitus and offspring's carotid intima-media thickness at birth: MySweetHeart Cohort Study
AUTHORS	Epure, Adina Mihaela; Di Bernardo, Stefano; Mivelaz, Yvan; Estoppey Younes, Sandrine; Chiolero, Arnaud; Sekarski, Nicole

VERSION 1 – REVIEW

REVIEWER	Baragetti, A Universita degli Studi di Milano, Dept. Pharmacological and Biomolecular Sciences
REVIEW RETURNED	30-Mar-2022

GENERAL COMMENTS	The work of the authors is valuable, considering the technical difficulty to measure this complex vascular parameter with an acceptable quality in so peculiar patients characteristic. Regarding the measurement of the IMT. Even though authors provide a well description on how the examination on th offsprings was evaluated, the distribution of IMT involves also subjects with 0.35 mm - 0.40 mm which is still a bit large for so pediatric subjects. How much do authors discuss on the spatial resolution of the probe that would affect the actual good parameter? According to previous point I suggest to change boxplots into dots plots graphs: this will allow to better appreciate the distribution of data. No information on the familial history of cardiovascular disease is provided: were there some mothers with still positive history of premature cardiovascular death? would do authors might imagine an inheritable connection with these findings? Table 2 is not so clear. Is it presenting an odds-ratio or a mean difference of IMT? in the latter case, which is the actual comparator? I think a graphical representation of this data (e.g.: forest plot) would be more informative.
---

VERSION 1 – AUTHOR RESPONSE

REVIEWER COMMENTS

Dr. A Baragetti, Università degli Studi di Milano, Ospedale Bassini

3. Comment: The work of the authors is valuable, considering the technical difficulty to measure this complex vascular parameter with an acceptable quality in so peculiar patient characteristic.

Response: We thank Dr Baragetti for this positive comment.

4. Comment: Regarding the measurement of the IMT. Even though authors provide a well description on how the examination on the offspring was evaluated, the distribution of IMT involves also subjects with 0.35 mm - 0.40 mm which is still a bit large for so pediatric subjects. a) How much do authors discuss on the spatial resolution of the probe that would affect the actual good parameter? b) According to previous point I suggest to change boxplots into dots plots graphs: this will allow to better appreciate the distribution of data.

Response: We agree that measurement error in CIMT due to limited spatial resolution cannot be excluded and we acknowledged this as a limitation in the discussion section: "Thirdly, CIMT was assessed using conventional high-resolution ultrasound frequencies (< 15 MHz), which have a lower spatial axial resolution and, thus, tend to overestimate the arterial thickness in the young children when compared to very high-resolution ultrasound systems (25 to 55 MHz).[30,31] Measurement error in CIMT cannot be excluded, but systematic differences between the two groups are unlikely because the outcome assessors were blinded to the glycemc status of the mothers."

b) We thank Dr Baragetti for this suggestion and we agree that the dot plots are also an effective way to show the spread of the data. However, in our case, the dot plot looks a bit crowded because of the sample size. We prefer therefore to have this figure in the supplemental material and keep in the paper the box plots due to their neater figure output. We provide additional details in the legend of Fig. 2 for easing the reading of the box plot: "The line inside the box represents the median value of the distribution, while the lower and upper boundaries of the box represent the first (Q1) and third quartiles (Q3), respectively. The interquartile range (IQR) corresponds to $Q3 - Q1$. The whiskers extend from either side of the box up to $1.5 \cdot IQR$ (ie, $Q1 - 1.5 \cdot IQR$ and $Q3 + 1.5 \cdot IQR$). Outliers are depicted as circles."

Fig. S2 Dot plot of CIMT at birth by gestational diabetes mellitus (GDM) and assignment to a lifestyle and psychosocial intervention (I).

Figure legend This figure shows the distribution of CIMT in the offspring of women without GDM (Non-GDM) and the offspring of women with GDM who were assigned to no intervention (GDM, non-I) or to a lifestyle and psychosocial intervention (GDM, I) as part of their participation in the MySweetHeart Trial. Abbreviations: CIMT, carotid intima-media thickness; GDM, gestational diabetes mellitus; I, intervention.

5. Comment: No information on the familial history of cardiovascular disease is provided: a) were there some mothers with still positive history of premature cardiovascular death? b) would do authors might imagine an inheritable connection with these findings?

Response: We agree that family history of premature cardiovascular death is a risk factor for early cardiovascular disease, (<https://pubmed.ncbi.nlm.nih.gov/22917005/>; <https://pubmed.ncbi.nlm.nih.gov/23578356/>; <https://www.ncbi.nlm.nih.gov/pmc/articles/PMC4536582/>), but we do not have the data to address its relationship with CIMT in newborns. We acknowledged this as a limitation of our paper and included the following sentence in the Discussion section: “Fourthly, while we adjusted for key confounders at the analysis stage, there is a possibility of bias due to unmeasured factors, such as family history of premature cardiovascular death, or residual confounding due to the relatively small sample size and imprecision in the measurement of confounder variables, especially in those self-reported.”

6. Comment: Table 2 is not so clear.
 - a) Is it presenting an odds-ratio or a mean difference of IMT? In the latter case, which is the actual comparator?
 - b) I think a graphical representation of this data (e.g.: forest plot) would be more informative.

Response: a) Table 2 is presenting a mean difference in CIMT. The comparator is the group without GDM (denoted as “ref” in the table; abbreviation “ref” explained as reference group in the table notes). b) We clarified Table 2 and, as recommended by Dr Baragetti, we also present the results graphically. The footnote of Table 2 or the plot area of Fig 3 and its legend make clear that we present mean differences in CIMT between offspring of women with and without GDM. In addition, for consistency matters, we also clarified Table S1 in supplementary material as it has a layout similar to Table 2 and constructed a second forest plot with the estimates presented in Table S1 (see Fig. S1 in supplementary material).

Fig. 3 Illustration of the relationship of GDM with offspring’s CIMT at birth through a forest plot.

Figure legend The boxes represent the mean differences in CIMT between offspring of women with and without GDM (i.e., GDM versus non-GDM). The horizontal lines represent the 95% CIs. The plot was constructed using regression estimates and models presented in Table 2. Model specification: Model 1 is unadjusted, while Models 2 and 3 are adjusted for various factors as described in the methods and footnote of Table 2. Abbreviations: CI, confidence interval; CIMT, carotid intima-media thickness; GDM, gestational diabetes mellitus.

Fig. S1 Illustration of the relationship of GDM and assignment or not to a lifestyle and psychosocial intervention with offspring’s CIMT through a forest plot.

Figure legend The circles represent mean differences in CIMT between offspring of women with GDM assigned to no intervention (GDM, non-I) and offspring of women without GDM (Non-GDM). The triangles represent mean differences in CIMT between offspring of women with GDM assigned to a lifestyle and psychosocial intervention (GDM, I) and offspring of women without GDM (Non-GDM). The horizontal lines represent the 95% CIs. The plot was constructed using regression estimates and models presented in Table S1. Model specification: Model 1 is unadjusted, while Models 2 and 3 are adjusted for various factors as described in the methods and footnote of Table S1. Abbreviations: CI, confidence interval; CIMT, carotid intima-media thickness; GDM, gestational diabetes; I, intervention.